# Applying linear programming in evaluating employees in higher education: A case study

**Radoslaw Ryńca, Yasmin Ziaeian***

Department of Management, Wroclaw University of Science and Technology, Wroclaw, Poland

These authors contributed equally to this work.
* yasmin.ziaeian@pwr.edu.pl

**Data Availability Statement:** All relevant data supporting the findings of this study are contained within the manuscript and some Data cannot be shared publicly due to confidentiality and privacy considerations.

## Abstract

In the past few decades, any type of organization, from factories to government organizations, the banking sector, or educational institutions concentrates on increasing profit margins. To achieve this, one of the key factors is to achieve maximum output with minimum resources (input). Therefore, having an optimal plan to apply the resources has become extremely important for organizations. One of the relevant resources is Human resources. This paper presents a mathematical model as an aid for optimizing human resources in the higher education sector. The model includes seven stages: 1) determining the availability of research and teaching staff, 2) evaluating the research and teaching staff from the perspective of different stakeholders, 3) determining the cost of the availability of research and teaching staff, 4) examining the motivated employees, 5) developing a staff ranking, 6) developing the mathematical model, and 7) implementing the mathematical model.

## Introduction

The 21st century is the time of developing a knowledge-based economy in which higher education plays a key role. Globalization, international competition, innovation, and technological advances have emphasized the importance of HRM for competitive advantage. [1]. Human capital is the most important sustainable competitive advantage in the organization during constant changes [2]. Studies show that human resources are even more relevant than financial and material sources [3]. The importance of human resource management is increasing because employees are considered a primary element in achieving competitive advantage [4]. These developments require effective human resource management across multiple areas, including job satisfaction, job engagement, and organizational performance [5–7]. Many organizations consider that positive job-related attitudes such as commitment and engagement are strategic elements for their competitive advantage [5, 8]. To gain a competitive advantage, organizations need people with high levels of energy, productivity, and commitment in their workplace [9], as employees with a positive attitude in their workplace can drive organizational success [10, 11]. In this regard, the skills, qualities, and abilities of human resources are very important to define employees' results such as work engagement and organizational engagement. The success of educational organizations fundamentally depends on the quality of human resources and the effective management of staff. Therefore, educational institutions

PLOS ONE | https://doi.org/10.1371/journal.pone.0310183   January 17, 2025
1 / 19

**Funding:** The author(s) received no specific funding for this work.

**Competing interests:** I have read the journal's policy and the authors of this manuscript have no competing interests to declare.

have to recruit and develop their staff and optimize their qualifications. Qualified and well-trained university staff have higher motivation and engagement in research and teaching [12]. In this paper, the parameter attempted to optimize is the human resource. At the institutional level, the appropriate implementation of such practices can enhance university performance. Some studies also have shown that staff has a crucial role to play in improving university rankings in areas such as academic research, University reputation, community development, and teaching quality. therefore, several educational institutions use human resource management to achieve high achievements, foster positive synergies, organizational engagement, and promote work engagement [12–18].

In another word, higher education sectors must anticipate market trends, and this is only possible with human resource allocation and management. Therefore, for a company or any high education sector is important to establish how it will develop its human resources and how they want to optimize and evaluate their staff.

## Literature review and problem statement

### Literature review

**The importance of employment management in higher education.** In general, Human resource management is a kind of system that has direct influence on the attitudes and behavior of employees in an organization [13]. It can be also considered as a philosophy that can encourage the employees to reach the organization's goals and can have impact to get the organizational success [19]. Different processes like recruitment, training, employee's engagement are the key of the HRM practices [20]. Additionally, efficiency and effectivity at work, great teamwork, satisfaction and high level of job security are also included to this practice and all these elements have an impact to have an effective HRM systems [21]. In the higher education sector, HRM practices focus on evaluating and enhancing the knowledge and abilities of human resources [22]. Many research is also showing that having a proper HRM system has direct impact on the performance of employees by influencing their attitudes and behavior, leading to increased levels of productivity and ultimately higher levels of organizational performance [23–26]. And if the mentioned practices won't have any impact on the behavior and attitude of the employees, whole HRM system cannot be successful strategy [27, 28].

Meanwhile, high commitment and positive attitude of employees will be very helpful to achieve their goals and tasks [29, 30].

**Motivation in higher education.** The range of literature about motivation is extensive and it is also presented in different ways like motivation in psychology, life sciences, education, or economics. According to literature review, some definitions are as followed:

- Motivation is a process that starts, encourages, and leads a goal-oriented behavior [31].

- Motivation is the amount of effort that a person is willing to invest in order to achieve a specific goal [32].

- Motivation can be also described as foundations of behavior [33] and can be a multi-faceted process, particularly the "internal-external" dimensions [34].

Motivation characterizes the willingness of a person that follows a certain activity to do something [35–38]. Properly, the motivation of university faculty can be determined as a process, that will cause to a goal-oriented behavior of the faculty members [39] and also can deal with different descriptions of faculty members' behavior for example regarding investing their energy in the preparation for their classes or engagement in their research, and willingness to help their students. The importance of using a theoretical framework that concentrates on the

quality of faculty motivation has also considered in studies of their intrinsic motivation, where studies typically report strong levels of intrinsic motivation with only small intra-individual differences [40, 41] In general, the performance and quality of higher education sectors like universities, colleges are vital to the society and the quality of these sectors are directly connected to the student's engagement, their learning outcome and their permanence [42–45]. High level of innovative research is produced in the university faculties and also, they can support a foster disciplinary advancement plus scientific progress [46]. Additionally, teaching and research in the faculties at the social level can be a basic element for the knowledge of the citizen, scientific development, economic enterprises and executive decision-making [47, 48]. Even with increasing the relevance of the faculty teaching and research, there are many troubles related to the faculty conditions and behaviors. For example, in many faculties, the number of published articles published in the major worldwide journals has fallen while spending in the research has increased [49]. And this is because that regularly, the importance of teaching is getting reduced time to time even by forcing the teaching institution to reach all research demands [50, 51].

**Evaluation discipline of employees in the higher education.** In many universities, there are evaluation systems. Basic assessment criteria some universities are as follows: Teaching activity and achievements (1–5 pts), Research activity and achievements (1–5 pts), Organizational activities and achievements in raising professional competence (1–5 pts). According to the requirements of the Higher Education and Science Act, the assessment also includes compliance with the provisions on copyright and related rights, industrial property and the mechanism of assessment by students and doctoral students (in the form of questionnaires). This information is not included in the score for the individual criteria, but is taken into account by the evaluation committee [52].

**Linear programming.** One of the techniques that examines the maximum or minimum linear objective function considering constraints to obtain optimal solution is the linear programming. Originally, this technique was developed before World War II in the military and industrial fields and later in other areas such as in financial, marketing and farming fields [53]. In this model, limitations and goals will be formulated as a linear function. This linear function includes three fundamental elements: decision variables, maximization or minimization and limitations. Decision variables are variables that affect the goal and determine the final function. The optimal solution will be reached with a combination of decision variables and their coefficients through this mathematical function [54].

- **Linear Programming: advantages and disadvantages**

Linear programming has been selected for this study because of the main following advantages:

- This technique is easy for decision-makers to use the resources effectively and they can get higher quality level of their decisions.

- With considering the limitations, this technique is very practical for the decision-makers to have an optimal solution.

- Identification of the constraints are most important advantage of this technique that could be used in evaluation process.

- Another advantage of this technique is quick adjustment if changes in conditions appear.

In another word, the advantages of linear programming in this project include ease of resource allocation, practical decision-making considering constraints, and quick adjustment

to changes. Disadvantages include the need for quantifiable objectives and measurable activities, which might not always be possible, and the requirement for viable alternative actions to determine resource limitations.

There are also some disadvantages to apply linear programming:

- Decision makers should be always able to formulate a quantitative objective and also all considered activities and resources, which are used to reach the goal, must be also measurable.

- The relationships regarding the objective function and the constraints must be linear in nature, which is not possible and also decision maker should have always viable alternative actions to determine the resources limitations [55, 56].

Despite the limitations of this technique, this research has been done with applying the linear programming to formulate the mathematical model and solve the problem.

**Ranking objects in the multi-criteria evaluations.** The literature on the subject offers numerous examples of the application of multi-criteria methods in the decision-making process. The ranking is used wherever resources are scarce to accomplish specific tasks. Therefore, it is necessary to rank objects according to their importance. One of the most used methods is object ranking [57–59]. In many branches of science and management area, this tool has been applied. The clarity, simplicity, and proper functionality of this method are its advantages in the decision-making process. The Rankine process involves computing the objects O that are objects of ranking, and objects of ranking in a higher education sector can be strategic objectives

$$O = \left\{ O_1, O_{2,...,} O_r \right\}$$

Where r is the number of objects under study.

Each object under analysis is a set of diagnostic variables X that describe the phenomena in the object. In the case under discussion, the diagnostic variables may include timing and cost-, the assessment of access to the resources needed to implement the strategic objective.

Set of diagnostic variables

$$X = \left\{ X_1, X_{2,...,} X_s \right\}$$

where s is the number of diagnostic variables.

The foundation for developing a ranking of strategic goals is the division of the diagnostic variable X into three subsets. The first group, called stimulants, includes variables where an increase indicates a positive effect on the assessment of a complex phenomenon. The second group includes stimulants, for example, diagnostic variables whose increase should be related to a decrease in the assessment of the phenomenon under consideration [60].

The second group, called destimulants, includes variables where an increase indicates a negative effect on the assessment of the phenomenon. The last group, called nominants, includes variables that have a certain value most favorable from the point of view of evaluating a complex phenomenon [61]. In order to be able to carry out a multi-criteria evaluation of individual phenomena, it is necessary to transform the values of the original characteristics. This requires a normalization process consisting of the transformation of diagnostic variables that assume values of similar magnitude without changes [61].

Stimulants based on the formula have been standardized (1)

$$z_{ij} = \frac{x_{ij} - \min\limits_{i} x_{ij}}{\max\limits_{i} x_{ij} - \min\limits_{i} x_{ij}} \tag{1}$$

where: i = 1,2,3,...r and j = 1,2,3...,s

$$X_j \in S$$

S- a subset of diagnostic variables called stimulants.

The destimulants are normalized by using formula (2)

$$z_{ij} = \frac{\max_i x_{ij} - x_{ij}}{\max_i x_{ij} - \min_i x_{ij}} \tag{2}$$

where: i = 1,2,3,...r and j = 1,2,3...,s

$$X_j \in D$$

D- a subset of diagnostic variables called stimulants.

The nominees are normalized according to the nature the variables assume. When a nominate takes on a certain value $C_{oj}$

Following formula is used (3):

$$
\begin{cases}
\dfrac{x_{ij} - \min_i x_{ij}}{c_{0j} - \min_i x_{ij}}, & gdy\ x_{ij} < c_{0j}, \\
1 & gdy\ x_{ij} = c_{oj}, \quad X_j \in N, \\
\dfrac{x_{ij} - \max_i x_{ij}}{c_{0j} - \max_i x_{ij}} & gdy\ x_{ij} > c_{0j},
\end{cases}
\tag{3}
$$

If the nominant is a set of $< C_{1j}, C_{2j}>$ formula should be applied (4):

$$
\begin{cases}
\dfrac{x_{ij} - \min_i x_{ij}}{c_{1j} - \min_i x_{ij}}, & gdy\ x_{ij} < c_{1j}, \\
1 & gdy\ c_{1j} \le x_{ij} \le c_{2j}, \quad X_j \in N, \\
\dfrac{x_{ij} - \max_i x_{ij}}{c_{2j} - \max_i x_{ij}} & gdy\ x_{ij} > c_{2j}.
\end{cases}
\tag{4}
$$

The normalization method presented above is called the zero-unitization method. It allows the assumption of a fixed reference point at which the interval of the normalized quantity is constant and equal to 1, while the normal are from the interval <0,1> (Kukuła et al. 2009).

The result of the normalization is the matrix shown in Table 1.

The expansion of the ranking needs a calculation of aggregated (synthetic) variables based on a formula (5):

$$Q_i = \sum_{j=1}^{s} z_{ij} \ (i = 1, 2, 3 \ldots, r) \tag{5}$$

Where $Qi$ is a synthetic variable that is a multi-criteria assessment of a complex phenomenon that characterizes an i-th object. The higher the value of the synthetic variable $Qi$, the better the position of a given object in the ranking.

**Table 1. Stimulants for the analyzed diagnostic variables and the aggregate variable.**

|  | $Z_{i1}$ | $Z_{i2}$ | $Z_{i3}$ | $Z_{i4}$ | $Z_{i5}$ | $Z_{i6}$ | $Z_{i7}$ | $Z_{i8}$ | $Z_{i9}$ | Q |
|---|---|---|---|---|---|---|---|---|---|---|
| $P_1$ | 1,00 | 0,64 | 0,05 | 0,50 | 0,00 | 0,07 | 0,00 | 0,00 | 0,00 | 2,26 |
| $P_2$ | 0,57 | 0,45 | 0,47 | 0,17 | 0,01 | 0,93 | 0,00 | 0,00 | 0,00 | 2,6 |
| $P_3$ | 0,51 | 0,64 | 0,41 | 0,50 | 0,00 | 0,93 | 0,17 | 0,00 | 0,00 | 3,16 |
| $P_4$ | 0,37 | 0,55 | 0,41 | 0,33 | 0,01 | 0,76 | 0,33 | 0,13 | 0,25 | 3,14 |
| $P_5$ | 0,24 | 0,73 | 0,78 | 0,67 | 1,00 | 0,93 | 0,50 | 1,00 | 1,00 | 6,85 |
| $P_6$ | 0,53 | 1,00 | 0,32 | 1,00 | 0,00 | 0,21 | 0,50 | 0,00 | 0,00 | 3,56 |
| $P_7$ | 0,71 | 0,45 | 0,28 | 0,17 | 0,01 | 0,28 | 0,00 | 0,00 | 0,50 | 2,4 |
| $P_8$ | 0,14 | 1,00 | 0,57 | 1,00 | 0,04 | 1,00 | 0,00 | 0,07 | 0,00 | 3,82 |
| $P_9$ | 0,33 | 0,64 | 0,55 | 0,50 | 0,04 | 0,24 | 0,17 | 0,00 | 0,25 | 2,72 |
| $P_{10}$ | 0,16 | 0,55 | 0,96 | 0,17 | 0,01 | 0,34 | 0,00 | 0,00 | 0,00 | 2,19 |
| $P_{11}$ | 0,29 | 0,36 | 0,80 | 0,67 | 0,01 | 0,21 | 0,00 | 0,87 | 0,00 | 3,21 |
| $P_{12}$ | 0,00 | 0,45 | 1,00 | 0,83 | 0,70 | 0,28 | 1,00 | 0,07 | 0,25 | 4,58 |
| $P_{13}$ | 0,57 | 0,55 | 0,12 | 0,00 | 0,00 | 0,00 | 0,00 | 0,13 | 0,25 | 1,62 |
| $P_{14}$ | 0,41 | 0,00 | 0,00 | 0,67 | 0,19 | 0,03 | 0,17 | 0,13 | 0,25 | 1,85 |
| $P_{15}$ | 0,27 | 0,64 | 0,88 | 0,83 | 0,03 | 0,38 | 0,33 | 0,00 | 0,00 | 3,36 |

Source: own elaboration

Z. Kukuła showed that also group objects can be distinguished as the best, average, and worst objects. (Kukuła et al. 2009).

For this aim, the formula can be used (6):

$$U = \frac{\max_i Q_i - \min_i Q_i}{3} \tag{6}$$

resulting in a subgroup of best objects for $Q_i \in (\max_i Q_i - U, \ \max_i Q_i >$

the subgroup of average objects for $Q_i \in (\max_i Q_i - 2U, \ \max_i Q_i - U >$

subgroup of the worst objects for $Q_i \in (\max_i Q_i, \ \max_i Q_i - 2U >$

The selection of this mathematical model is underpinned by its inherent utility, offering the distinct advantage of facilitating the generation of object rankings, enabling the categorization of objects into three distinct groups. Noteworthy is its enduring applicability and user-friendly nature, as it obviates the necessity for the deployment of sophisticated software tools, with Microsoft Excel alone serving as a sufficient platform for data manipulation and analysis.

## Problem statement

The importance of human resource management and their performance are already realized in the higher education sectors to fulfill the expectations and needs of their students. In this manner, it is relevant to increase student satisfaction because colleges, private and public universities are expanding quickly and also education fees are rising [62] and the education market became very competitive. In such a situation and competitive market, it's important to manage the human resources and optimize their performance in strategic way. Already, many authors recognized, how important is human resource management, evaluating, defining the motivation of employees in higher education institutions, ranking system and managing lectures [63–71], but it is necessary to apply an effective model and optimizing the related criteria.

Based on problem statement, this study focuses on the important question, how high education sector can optimize their performance of their employees to get an effective result in a very competitive market and the aim of this research is presenting a mathematical model to aid optimizing the human resources.

Linear programming is used in this study because the relationships and constraints involved in the problem are linear. The objective functions and constraints can be represented as linear equations, making linear programming a suitable and effective method for optimization in this context.

## Research gap

Many researchers and authors have addressed aspects of human resources management in higher education [12, 72–75] or there is also study like Evaluating University Reputation Based on Integral Linear Programming [64], but the research on engagement in higher institutions is very limited [76–78] and other related issues such as engagement are not well researched in higher education [78]. In general, previous research in higher education has focused more on issues such as faculty and administrative morale [79], staff job satisfaction [80], and professors' intention to leave [81]. And in general, the research gaps show there is any mathematical model that was applied specifically for the human resource management at universities. According to this model, there is some research about Academic Plans Design, Resource planning at universities and Strategy of University image [82–85].

## Methodology

Fig 1 shows the implementation of the proposed model:

*The first step (stage 1.)* of the proposed model includes determining **the availability of research and teaching staff** ready to teach in the new field of study. The objective of Stage 1 is to assess the availability and potential involvement of current research and teaching staff in the new program. This stage aims to gather detailed information about the staff's current commitments, skills, experience, and potential availability for new assignments. This stage should be preceded by an analysis of the strengths and weaknesses of its own potential, which are the employees of the university. The aim of this stage is also to obtain information on the teaching staff's salary. The employees are often involved in the teaching process in various fields of study, which may hinder the planning of teaching and consequently their involvement in the newly created field of study.

Offering a new, elite course of study and bearing in mind the high level of education and research conducted, the management may face the problem of deciding which employees to engage first, bearing in mind their competencies, experience, and availability for the new course opening (expressed in the "free" teaching salary). Generally, following points will be considered in this stage:

- Staff Audit: Collect data on current teaching loads, research commitments, and other obligations. Create a database with each staff member's details, including workload.

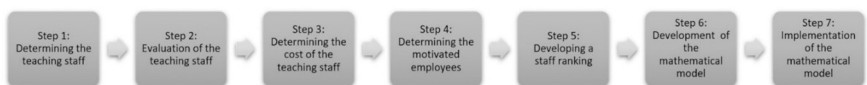

**Fig 1. The model implementation process.** Source: Own study.

- Skills and Competency Mapping: Conduct a skills inventory and assess competencies relative to the new program's requirements.

- Availability Analysis: Analyse current workloads to determine staff availability and calculate available hours.

- Performance Review: Review historical performance based on evaluations, research outputs, and feedback from students, peers, and management.

- Competency Match: Match skills and competencies with program needs, considering inter-disciplinary potential.

- Mitigation of Barriers: Identify and mitigate potential barriers such as ongoing projects or personal commitments.

- Selection and Documentation: Select suitable staff for the new program and document the process, including data and rationale.

The output of Stage 1 is a detailed report on staff availability and competencies, a list of selected staff members with roles and responsibilities, and documentation of the analysis and decision-making process.

***Step 2****. Evaluation of the research and teaching staff from the perspective of different stakeholders*

In the proposed model, the authors recommend evaluating employees from the perspective of students and university management. For evaluation from the students' perspective, it is necessary to develop a survey questionnaire with a set of evaluation criteria and conduct a survey. Each criterion would be rated on a scale of 1 to 5, where 1 is rated very low, 5- is rated very high. Examples of criteria are shown in Table 2.

The evaluation of the scientific and teaching staff from the perspective of the university management is also important. In particular, the degree of involvement of the staff in the conducted scientific research during the evaluation period for a given scientific discipline.

Examples of evaluation criteria could be:

**Table 2. Example of criterion evaluation.**

|  | CRITERION EVALUATION | | | | |
|---|---|---|---|---|---|
|  | **1** | **2** | **3** | **4** | **5** |
| Professional level of the academic teacher |  |  |  |  |  |
| Involvement of academic staff in conducting classes |  |  |  |  |  |
| Availability of academic staff during consultations |  |  |  |  |  |
| Evaluation of the sharing of knowledge from the academic teacher with students |  |  |  |  |  |
| Evaluation of the university teacher's practical experience in working with students |  |  |  |  |  |
| Academic support for students in their academic development |  |  |  |  |  |
| A fair policy for assessing student work |  |  |  |  |  |
| Communication competences of academic staff |  |  |  |  |  |
| The content of lectures is well connected with the content of other classes forming the subject |  |  |  |  |  |
| Ethical competence (e.g., responding to unethical behavior) of the university teacher |  |  |  |  |  |

Source: Own elaboration

- the number of publication credits earned

- number of publications in renowned scientific journals

- the number of citations of a member of the university staff

- number of PhDs promoted by the staff member

- number of research grants and projects carried out by the employee

- number of patents and new inventions obtained by the employee

- the number of awards, certificates, distinctions, and qualifications obtained by the employee.

**Step 3. *Determining the cost of availability of research and teaching staff***

Taking into account the financial constraints related to the access to highly qualified research and teaching staff and the budget provided for the implementation of the new field of study, it is necessary to determine the costs of resource availability. This stage may consist in identifying the salary costs of employees, proportional to their involvement in the new field of study.

**Step 4. *Determining the motivated employees***

Properly motivated staff may be a key factor influencing the quality of the teaching process and the conducted research. We propose that this stage should be carried out on the basis of a survey questionnaire and a survey conducted on a group of university employees. The aim of the survey would be to assess the degree of motivation of the research and teaching staff, carried out on the basis of a number of evaluation criteria. Each criterion would be rated by employees on a scale of 1 to 5, where 1 would be rated very low, 5- rated very high. Examples of evaluation criteria are shown in Table 3.

**Step 5.: *Developing a staff ranking***

While selecting employees, the following factors may be important: a) the cost of the employee's involvement in the teaching process in the new faculty (measured by the average size of the remuneration for the position, proportional to the involvement in the new faculty), b) its importance from the perspective of various stakeholders (e.g., high teaching evaluation of the employee—which may be important from the perspective of the students, and the

**Table 3. Example of evaluation criteria.**

| | CRITERION EVALUATION | | | | |
|---|---|---|---|---|---|
| | 1 | 2 | 3 | 4 | 5 |
| How would you rate your supervisor's management competencies? | | | | | |
| How would you rate the safety of your working conditions? | | | | | |
| How would you rate the degree to which the university's infrastructure is adapted to your needs? | | | | | |
| How do you assess the fairness of the employee evaluation system? | | | | | |
| How secure do you feel in terms of permanent employment? | | | | | |
| How closely does the scope of your work and responsibilities match your salary? | | | | | |
| How would you rate the fairness of rewarding and valuing your work? | | | | | |
| How would you rate the additional non-wage benefits available? | | | | | |
| How motivated are you to work by the incentive system in place? | | | | | |
| How would you rate your career opportunities? | | | | | |

Source: Own elaboration

number of points awarded for publications or the number of promoted doctorates—important from the perspective of the university management, taking into account the maintenance of the proper evaluation policy of the scientific discipline. It may also be important to c) the availability of the employee for the new faculty (measured by the amount of free teaching salary possible to engage in the new faculty, or d) assess the employee's degree of motivation to work.

The university management may therefore be faced with the complex problem of deciding which employees to engage first, taking into account various selection criteria. The method of ranking objects in the light of multi-criteria evaluations may be helpful in the discussed scope. The proposed ranking makes it possible to indicate which facilities (employees) are key from the evaluators' point of view and which should be involved in the teaching and research process in the new field of study.

The authors propose in this stage of implementation of the proposed model (stage 5 to use the method of ranking according to the procedure presented in K. Kukuła1. The result of using this procedure is a ranking of research and teaching staff in the form of three groups. For further analysis, the authors of the article propose to include only those in the group of best employees.

***Stage 6***. ***Development of the mathematical model.***

In this stage of the proposed methodology, evolving the mathematical model is a crucial step that encapsulates optimizing human resources in the higher education sector. This stage involves developing a mathematical model in the form of an objective function as well as constraints. In this stage of the proposed methodology, developing the mathematical model is a crucial step that encapsulates optimizing human resources in the higher education sector. The mathematical model is formulated by integrating the objective function and the constraints. This model is then solved using linear programming techniques to find the optimal solution. The optimal solution identifies which employees should be involved in the new program and to what extent, ensuring maximum effectiveness while adhering to the constraints.

***Stage 7***. ***Implementation of the mathematical model into the computer environment.***

Currently, there are many programs on the market that allow to solve linear programming problems. Examples include Solver, LindoApiSystem, Gusek.

## The use of linear programming in human resource policy management in a new field of study—A proposed model

The implementation of a new faculty at a department requires the use of appropriate tools for its implementation. According to the authors of the article, it may be justified to use linear programming in managing the human resources policy at the new faculty, taking into account its elite character and high level of education and scientific research. As it was mentioned before, the selection of research and teaching staff to work at the new faculty may cause problems, which staff to engage first, bearing in mind the greatest satisfaction of various stakeholders (management, students), their availability measured by the possibility to engage in the research and teaching process at the new faculty, while taking into account the financial resources available to the university.

Bearing in mind that the management of the university should strive for a situation in which it is possible to satisfy the needs of the various stakeholders as much as possible, the criterion for the selection of staff for the new course of study takes the form of:

$$\sum_{i=1}^{n} (d_i * t_i) * W_i \to \max$$

$$\text{Sum}\,(E_i * d_i * W_i) \to \text{MAX}$$

Where:

$d_i$—the decision variable in the form of commitment of the i-th employee to the teaching process in the new direction, where [$d_i$ 0,1].

$E_i$—evaluation of the i-th employee from the perspective of the students expressed as an average of all students

$W_i$—the evaluation of the i-th employee from the perspective of the university management expressed by the number of points awarded for publications in a given evaluation period.

Taking into account the limited possibility for staff to engage in the teaching process in a new field of study (as staff often teach in different fields of study) as a result of the fixed teaching salary and the policy of not using overtime beyond the fixed salary, the limitation of the presented model is as follows:

$$\sum_{i=1}^{n} di\,Zi \leq T$$

$$\text{Sum}\,(D_i * P_i) <= K$$

Where:

$P_i$—the number of hours of the teaching load of the i-th employee that can be used in the new field of study

$K$—total number of teaching hours to be completed by staff in a given field of study

$d_i$—the decision variable in the form of commitment of the i-th employee to the teaching process in the new direction, where [$d_i$ 0,1]

It is also important to select staff in such a way as to take account of the financial constraints on the teaching process in the new course.

$$\text{Sum}\,(D_i * S_i) <= B$$

$S_i$—the amount of salary of the i-th employee, proportional to his/her involvement in the teaching process in the new field of study

$B$—the size of the financial budget of the higher education institution foreseen for the implementation of the new field of study

$d_i$—the decision variable in the form of commitment of the i-th employee to the teaching process in the new direction, where [$d_i$ 0,1]

Conducting an appropriate human resources policy is not an easy task. It requires the university management to take into account various variables and evaluation criteria. Research and teaching staff may not have the same importance from the university management's perspective and may have different impacts on satisfaction from the students' perspective. It is therefore necessary to select them in an optimal way, taking into account their availability in the commitment to teaching and research activities in the new field of study and the financial capabilities of the university. We believe that it may be helpful in this case to use linear programming in order to find an optimal solution given the existing constraints. As mentioned earlier linear programming is used in the situation of a decision problem where the intention of the decision maker is to find an optimal solution in the form of a maximum or minimum. The authors of the article point out a number of advantages of using the proposed method. It can be helpful in the process of personnel planning for a new field of study, in particular, taking into account the needs and expectations of various university stakeholders. Linear

programming can provide information on the optimal selection of research and teaching staff, given the existing limitations, e.g., resulting from access to time or financial resources. It also allows for evaluation of the staff taking into account their assessment from the point of view of different stakeholders and evaluation criteria, in a comprehensive manner with the occurring constraints. The authors of this paper are also aware of the limitations of the proposed model. The proposed model may be time-consuming to develop. Difficulties may also result from the lack of knowledge of managers concerning the development of the mathematical model, which may make it difficult or impossible to apply. The proposed solution also requires the use of IT tools for calculations (e.g., AMPL, SOLVER, GUSEK, STORM or others).

## Case study

The study was conducted at a university in Lower Silesia. 15 research, teaching and teaching staff participated in the study. The first stage consisted in determining the teaching availability of employees by determining their free teaching load that could be used in a new field of study (stage 1 of the model). Table 3 shows the results obtained. For example, a $P_1$ employee may be fully involved in the teaching process in a new field of study (300 hours -100% of the teaching load for a given position). Laboratory $P_7$ can carry out 225 hours. in a new field, while employee $P_{12}$ only 45 hours.

Employees were also assessed from the perspective of various stakeholders (stage 2 of the model). For this purpose, a survey was carried out on a group of 30 students using a questionnaire. The model of the questionnaire is presented in Table 4. A particular criterion would be assessed on a scale of 1 to 5, where 1—very low, 5—very high. Table 5 shows the average of the assessed criteria for individual employees. For example, the employee P6 was assessed the best from the students' perspective, and the $P_{14}$ employee the worst.

Employees were also assessed from the perspective of the university management. The following criteria were taken into account in the study: the total number of citations, the number of publications in scientific journals in a given period of evaluation of the scientific discipline, the number of Impact Factor publications, the number of doctors promoted by the employee, and the number of grants and research projects implemented by the employee. Detailed results are presented in Table 5. For example, employee P6, during the discipline evaluation period, wrote 7 publications, including 3 with Impact Factor. It also has a cited rate of level 3.

At a further stage of the research, the costs of the availability of research and teaching staff were determined (stage 3 of the model). It was determined on the basis of the average remuneration for a given position, in proportion to the free teaching load. For example, a $P_3$ employee working as an assistant professor with a workload of 225 hours. per year, they can only engage in a new field of study in less than 62% (140 hours). The value of PLN 46,200 represents 62% of his full annual salary.

**Table 4. Matrix of normalized variables.**

| Object i-th (Strategic objectives) | Diagnostic variables | | | |
|---|---|---|---|---|
| | **Xi1** | **Xi2** | . . . | **xij** |
| C1 | $z_{11}$ | $z_{12}$ | . . . | $z_{1j}$ |
| C2 | $z_{21}$ | $z_{22}$ | . . . | $z_{2j}$ |
| . . . | . . . | . . . | . . . | . . . |
| Ci | $z_{i1}$ | $z_{i2}$ | . . . | $z_{ij}$ |

Source: Own elaboration.

**Table 5. Result of employee's assessment.**

| Employees | Teaching salary to be used for new faculty (Number of hours) | Employee evaluation from the perspective of students (Grade point average) | Employee availability costs per annum (in PLN) | Degree of staff motivation (average score) | Number of points for publications | Number of publications in reputable scientific journals | Number of citations of a university employee | Number of PhDs promoted by the employee | Number of research grants and projects carried out by the employee |
|---|---|---|---|---|---|---|---|---|---|
| $P_1$ | 300 | 4,00 | 67 200 | 4,00 | 0 | 3 | 0 | 0 | 0 |
| $P_2$ | 190 | 3,60 | 42 660 | 3,50 | 6 | 28 | 0 | 0 | 0 |
| $P_3$ | 175 | 4,00 | 46 200 | 4,00 | 0 | 28 | 1 | 0 | 0 |
| $P_4$ | 140 | 3,80 | 46 128 | 3,75 | 4 | 23 | 2 | 2 | 1 |
| $P_5$ | 105 | 4,20 | 24 840 | 4,25 | 611 | 28 | 3 | 15 | 4 |
| $P_6$ | 180 | 4,80 | 51 360 | 4,75 | 3 | 7 | 3 | 0 | 0 |
| $P_7$ | 225 | 3,60 | 54 000 | 3,50 | 7 | 9 | 0 | 0 | 2 |
| $P_8$ | 80 | 4,80 | 36 792 | 4,75 | 24 | 30 | 0 | 1 | 0 |
| $P_9$ | 130 | 4,00 | 38 304 | 4,00 | 24 | 8 | 1 | 0 | 1 |
| $P_{10}$ | 85 | 3,80 | 14 112 | 3,50 | 5 | 11 | 0 | 0 | 0 |
| $P_{11}$ | 120 | 3,40 | 23 520 | 4,25 | 4 | 7 | 0 | 13 | 0 |
| $P_{12}$ | 45 | 3,60 | 12 000 | 4,50 | 429 | 9 | 6 | 1 | 1 |
| $P_{13}$ | 190 | 3,80 | 63 504 | 3,25 | 3 | 1 | 0 | 2 | 1 |
| $P_{14}$ | 150 | 2,60 | 70 200 | 4,25 | 117 | 2 | 1 | 2 | 1 |
| $P_{15}$ | 115 | 4,00 | 18 696 | 4,50 | 17 | 12 | 2 | 0 | 0 |

Source: Own elaboration

An important stage of the research was to determine the level of employee motivation (stage 4 of the model).

This stage was carried out on the basis of a questionnaire survey with the use of a questionnaire. The aim of the study would be to assess the level of motivation of the research and teaching worker based on a number of criteria, which are shown in Table 2. A particular criterion would be assessed by employees on a scale of 1 to 5, where 1 is very low, 5 is very high. The average results are presented in Table 5. For example, the most motivated employees are $P_6$ and $P_8$, while the least motivated are $P_{13}$.

The next stage (stage 5 of the model) was to develop a ranking of employees based on variables that are important both from the perspective of students and university management. For this purpose, the ranking of objects according to K. Kukuła was used.

The analyzed diagnostic variables were:

$X_1$—Teaching workload to be used in a new field of study

$X_2$—Employee evaluation from the students' perspective

$X_3$—Costs of employee availability on an annual basis (in PLN)

$X_4$—Degree of employee motivation

$X_5$—Total number of citations

$X_6$—Total number of publications in scientific journals

$X_7$—Total number of Impact Factor publications

$X_8$—The total number of doctors promoted by the employee

$X_9$—Total number of grants and research projects carried out by the employee

In order to determine the ranking, stimulants were determined for the diagnostic variable $X_3$, and stimulants for the variables $X_1$, $X_2$, $X_4$, $X_5$, $X_6$, $X_7$, $X_8$, $X_9$. The aggregate variable Q was also determined. The results are shown in Table 1.

Knowing the Q variable allows to build a ranking of employees ordered in relation to the non-increasing $Q_i$ values.

As a result of the conducted analysis, 3 subgroups of employees were obtained.

The result of the best workers: $P_5$

Average employee group: $P_6$, $P_8$, $P_{12}$, $P_{15}$.

Worst employee group: $P_1$, $P_2$, $P_3$, $P_4$, $P_7$, $P_8$, $P_9$, $P_{11}$, $P_{13}$, $P_{14}$.

Employees from the first two groups were taken for further analysis.

As already mentioned, the university management should strive to meet the needs of various entities as much as possible, the criterion for selecting employees for a new field of study takes the form:

$$\sum_{i=1}^{n} (d_i * t_i) * W_i \rightarrow \max$$

$$4,20*611*d_5 + 4.80*3*d_6 + 4.8*24*d_8 + 3.6*429*d_{12} + 4*17*d_{15} \rightarrow \text{MAX}$$

Taking into account the limited possibility of employees' involvement in the didactic process in the new field of study, the total number of hours envisaged by employees in a given field of study was 2,230 hours and the annual budget 0f 80,000 PLN.

The limitations of the presented model take the form of:

$$\text{Sum}\,(D_i * P_i) <= \text{K}$$

$$105*d_5 + 180*d_6 + 80d_8 + 45*d_{12} + 115*d_{15} <= 2230$$

$$\text{Sum}\,(D_i * S_i) <= \text{B}$$

$$24840*d_5 + 51360*d_6 + 36792*d_8 + 12000*d_{12} + 118696*d_{15} <= 80000$$

## Results and discussion

As mentioned, Linear Programming (LP) is one of the greatest ways to perform optimization and it helps solving very complex optimization problems by making assumptions. It is a method for calculating best outcome with a range of constraints, that are defined mathematically [86]. For this aim, AMPL is applied in this study as the software. AMPL (A Mathematical Programming Language) is an algebraic modeling language for explaining and finding out highly complex problems for large-scale mathematical calculations [87]. The model on AMPL and the optimal solution is determined and is shown in Fig 2:

The linear programming model is developed from the perspective of the decision-maker at university. Regarding the model, five employees have been considered and are more effective employee among others with consideration of the budget of university, cost hour of the employee, evaluation of students and publication.

```
option solver cplex;
reset;

var d5 binary;
var d6 binary;
var d8 binary;
var d12 binary;
var d15 binary;

maximize goal: 2566.2*d5+14.4*d6+115.2*d8+1544.4*d12+68*d15;

subject to

c1: 105*d5+180*d6+80*d8+45*d12+115*d15<=2230;

c2: 24840*d5+51360*d6+36792*d8+12000*d12+118696*d15<=80000;

solve;
display d5,d6,d8,d12,d15;
```

**Fig 2. The formulated model in AMPL.** Source: Own elaboration.

As it is shown in Fig 3, according to the solution, three employees will be selected for the new faculty (d5, d8 and d12) and two other employees, d8 and d15 won't be considered anymore.

## Limitation

This study has potential limitations. To apply all mentioned methods, more time and thought is required in the construction of the final model. The authors of the article are also aware of the weaknesses of the use of linear programming in such cases. This method can be time consuming to implement. It requires knowledge of the costs of teaching process and promotional activities and access to software in order to make calculations. It also requires skills related to the development of a mathematical model. In another words, self-reporting has its drawbacks, such as bias and inaccuracies. These limitations should be acknowledged, and measures should be taken to mitigate their impact. The selection of criteria is crucial, as what is important to management might be irrelevant to students. Customizing the model to include diverse perspectives can improve its applicability.

```
ampl: model 'C:\Users\YASMIN ZIAEIAN\linprog4.mod';
CPLEX 20.1.0.0: optimal integer solution; objective 4225.8
0 MIP simplex iterations
0 branch-and-bound nodes
d5 = 1
d6 = 0
d8 = 1
d12 = 1
d15 = 0
```

**Fig 3. The optimal solution for the objectives assumed in AMPL.** Source: Own elaboration.

## Conclusion

As the evaluation of employees and selecting them for new faculty is relevant, the study shows the applying of linear programming in human resource policy management in a new field of study. The linear programming model is also flexible and new constraints or goals can be easily added or changed to test the results. The proposed methodology can be very useful for other high education sectors, and it is very conceivable that some of the goals and constraints may not be taken into account. The presented proposal of using linear programming can be widely used in a university. It allows for the selection of the optimal solution under the existing assumptions. As a summary, it can be mentioned that the novelty of this research is the combination of different methods in the high education sector and create a new model to evaluate the employees in higher education sector in an effective way. As further research, this model can be applied for different level of strategic management in higher education sectors in any kind of decision-making process and evaluations.

## Author Contributions

**Data curation:** Radoslaw Ryńca.

**Methodology:** Radoslaw Ryńca, Yasmin Ziaeian.

**Resources:** Radoslaw Ryńca, Yasmin Ziaeian.

**Writing – original draft:** Yasmin Ziaeian.

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
