## [Decision Letter · Decision Letter 0]

2 Jul 2024

PONE-D-24-15475Applying the linear programming in the evaluation of employees in the higher education sector-Case studyPLOS ONE

Dear Dr. Ziaeian,

Thank you for submitting your manuscript to PLOS ONE. After careful consideration, we feel that it has merit but does not fully meet PLOS ONE’s publication criteria as it currently stands. Therefore, we invite you to submit a revised version of the manuscript that addresses the points raised during the review process.

We look forward to receiving your revised manuscript.

Kind regards,

Claudia Noemi González Brambila, Ph.D.

Academic Editor

PLOS ONE

Additional Editor Comments (if provided):

Reviewers' comments:

Reviewer's Responses to Questions

**Comments to the Author**

1. Is the manuscript technically sound, and do the data support the conclusions?

Reviewer #1: Partly

Reviewer #2: Yes

2. Has the statistical analysis been performed appropriately and rigorously? 

Reviewer #1: No

Reviewer #2: Yes

3. Have the authors made all data underlying the findings in their manuscript fully available?

Reviewer #1: Yes

Reviewer #2: Yes

4. Is the manuscript presented in an intelligible fashion and written in standard English?

Reviewer #1: Yes

Reviewer #2: Yes

5. Review Comments to the Author

Reviewer #1: I consider that the topic is interesting and useful.

It is recommended to enumerate all 7 stages, not just 5.

There is no justification or demonstration of the linearity of the problem to be solved, yet the entire argument relies on this assumption. Both advantages and disadvantages should be specifically related to the project.

The problem statement needs to be more specific.

At the beginning of the paper, the problem is not clearly stated; specifically, it is not clear that there are no new professors, and that they are to be selected for a new program.

Diagnostic variables are divided into 3 subsets, and 2 of them have the same name.

Stage 1 is weak and Stage 6 is not explained.

Table 4 is used as if it were Table 3 in the corresponding section.

The title of the paper does not correspond to the work carried out, as there is no comparison with an alternative methodology. Therefore, it cannot be concluded that costs were effectively reduced and benefits increased.

Reviewer #2: Page 2, second paragraph, first sentence. Change to higher education. Page 2, middle third paragraph. "Once, is related...." I don't understand that sentence. Also, "Many research...." probably should be much research. Page 3, Last paragraph, second sentence, the end should be "are as follows:" You present an interesting tool for employee evaluation. As I understand it, surveys are used to collect data. Self-reporting has its drawback. Do you want to address it as part of your limitations? The selection of criteria I o important. What may be important to management is irrelevant to the students. The model does seem to be customizable. The biggest drawback I see I the self-reporting survey data and elected variables which could make results not vary useful. It would depend on the circumstances. Good luck with this model.

6. PLOS authors have the option to publish the peer review history of their article (what does this mean?). If published, this will include your full peer review and any attached files.

Reviewer #1: No

Reviewer #2: No

---

## [Author Response · Author response to Decision Letter 0]

7 Aug 2024

Dear Madam/Sir

Thank you for your constructive feedback on our manuscript. We have carefully considered the comments from both reviewers and made the necessary revisions to address each point. The revised manuscript has already been submitted. Below is a detailed summary of the changes made in response to the reviewers' comments:

Reviewer #1 Comments:

Enumerate all 7 stages, not just 5.

We have revised the manuscript to explicitly enumerate all seven stages of the proposed model in the methodology section.

Justification or demonstration of the linearity of the problem.

Added a justification for the linearity of the problem in the methodology section, explaining why linear programming is appropriate for this study.

Specific advantages and disadvantages related to the project.

Included a discussion on how the advantages and disadvantages of linear programming specifically relate to the project.

Clear problem statement.

Clarified the problem statement at the beginning of the paper, emphasizing that no new professors are being hired and the selection is for a new program.

Diagnostic variables.

Corrected the naming and categorization of diagnostic variables on Page 5 to ensure no duplicates. The three subsets are now clearly distinguished as stimulants, destimulants, and nominants.

Weak Stage 1 and unexplained Stage 6.

Expanded Stage 1 to include detailed steps involved in determining the availability of research and teaching staff. Provided a clearer explanation of Stage 6, detailing the development of the mathematical model.

Correction of Table references.

Checked and corrected all table references to ensure they are correctly used in the corresponding sections.

Title of the paper.

Modified the title to better reflect the content: “Applying Linear Programming in Evaluating Employees in Higher Education: A Case Study”.

Reviewer #2 Comments:

Page 2, second paragraph, first sentence. Change to higher education.

Revised the sentence to: “In the higher education sector, HRM practices focus on evaluating and enhancing the knowledge and abilities of human resources.”

Page 2, middle third paragraph. "Once, is related...." I don't understand that sentence. Also, "Many research...." probably should be much research.

Clarified and revised the sentence to: “In the higher education sector, HRM practices focus on evaluating and enhancing the knowledge and abilities of human resources. Much research shows that having a proper HRM system directly impacts employee performance by influencing their attitudes and behavior, leading to increased productivity and higher organizational performance.”

Page 3, Last paragraph, second sentence, the end should be "are as follows:"

Corrected the sentence to end with “are as follows:”.

Address self-reporting limitations and selection of criteria.

Added a significant limitation related to the reliance on self-reporting survey data to the limitations section. Discussed the importance of selecting criteria that align with both management and student priorities, and highlighted the model's customization potential.

We believe these revisions have significantly improved the manuscript and addressed the reviewers’ concerns.

 Thank you for the opportunity to revise our work. We look forward to your feedback.

Sincerely,

Radoslaw Ryńca and Yasmin Ziaeian

Department of Organization and Management

Faculty of Management

Wroclaw University of Science and Technology

---

## [Decision Letter · Decision Letter 1]

27 Aug 2024

Applying the linear programming in the evaluation of employees in the higher education sector-Case study

PONE-D-24-15475R1

Dear Dr. Ziaeian,

We’re pleased to inform you that your manuscript has been judged scientifically suitable for publication and will be formally accepted for publication once it meets all outstanding technical requirements.

Kind regards,

Claudia Noemi González Brambila, Ph.D.

Academic Editor

PLOS ONE

Additional Editor Comments (optional):

Reviewers' comments:

Reviewer's Responses to Questions

**Comments to the Author**

1. If the authors have adequately addressed your comments raised in a previous round of review and you feel that this manuscript is now acceptable for publication, you may indicate that here to bypass the “Comments to the Author” section, enter your conflict of interest statement in the “Confidential to Editor” section, and submit your "Accept" recommendation.

Reviewer #1: All comments have been addressed

Reviewer #2: All comments have been addressed

2. Is the manuscript technically sound, and do the data support the conclusions?

Reviewer #1: Yes

Reviewer #2: (No Response)

3. Has the statistical analysis been performed appropriately and rigorously? 

Reviewer #1: Yes

Reviewer #2: (No Response)

4. Have the authors made all data underlying the findings in their manuscript fully available?

Reviewer #1: Yes

Reviewer #2: (No Response)

5. Is the manuscript presented in an intelligible fashion and written in standard English?

Reviewer #1: Yes

Reviewer #2: (No Response)

6. Review Comments to the Author

Reviewer #1: After the authors have addressed the previuos observations, I consider the paper suitable for publication. It only requires a final revision for minior formating issues.The work is relevant to the field of higher education.

Reviewer #2: (No Response)

7. PLOS authors have the option to publish the peer review history of their article (what does this mean?). If published, this will include your full peer review and any attached files.

Reviewer #1: No

Reviewer #2: No

---

## [Editor Report · Acceptance letter]

9 Sep 2024

PONE-D-24-15475R1 

PLOS ONE

Dear Dr. Ziaeian, 

I'm pleased to inform you that your manuscript has been deemed suitable for publication in PLOS ONE. Congratulations! Your manuscript is now being handed over to our production team.

Kind regards, 

on behalf of

Dr. Claudia Noemi González Brambila 

Academic Editor

PLOS ONE